# Caching Policy in Low Earth Orbit Satellite Mega-Constellation Information-Centric Networking for Internet of Things

**DOI:** 10.3390/s24113412

**Published:** 2024-05-25

**Authors:** Hongqiu Luo, Tingting Yan, Shengbo Hu

**Affiliations:** 1Institute of Intelligent Information Processing, Guizhou Normal University, Guiyang 550001, China; luohqiu@163.com (H.L.); yantingting@gznu.edu.cn (T.Y.); 2School of Computer and Electronic Information, Guizhou Qiannan College of Science and Technology, Guiyang 550600, China; 3Center for RFID and WSN Engineering, Department of Education Guizhou, Guizhou Normal University, Guiyang 550001, China; 4China and National Space Science Center, Chinese Academy of Sciences (CAS), Beijing 100190, China

**Keywords:** low earth orbit (LEO) satellite mega-constellation, information-centric network, caching policy, random forest, network simulator 3 (NS-3)

## Abstract

Information-Centric Networking (ICN) is the emerging next-generation internet paradigm. The Low Earth Orbit (LEO) satellite mega-constellation based on ICN can achieve seamless global coverage and provide excellent support for Internet of Things (IoT) services. Additionally, in-network caching, typically characteristic of ICN, plays a paramount role in network performance. Therefore, the in-network caching policy is one of the hotspot problems. Especially, compared to caching traditional internet content, in-networking caching IoT content is more challenging, since the IoT content lifetime is small and transient. In this paper, firstly, the framework of the LEO satellite mega-constellation Information-Centric Networking for IoT (LEO-SMC-ICN-IoT) is proposed. Then, introducing the concept of “viscosity”, the proposed Caching Algorithm based on the Random Forest (CARF) policy of satellite nodes combines both content popularity prediction and satellite nodes location prediction, for achieving good cache matching between the satellite nodes and content. And using the matching rule, the Random Forest (RF) algorithm is adopted to predict the matching relationship among satellite nodes and content for guiding the deployment of caches. Especially, the content is cached in advance at the future satellite to maintain communication with the current ground segment at the time of satellite switchover. Additionally, the policy considers both the IoT content lifetime and the freshness. Finally, a simulation platform with LEO satellite mega-constellation based on ICN is developed in Network Simulator 3 (NS-3). The simulation results show that the proposed caching policy compared with the Leave Copy Everywhere (LCE), the opportunistic (OPP), the Leave Copy down (LCD), and the probabilistic algorithm which caches each content with probability 0.5 (prob 0.5) yield a significant performance improvement, such as the average number of hops, i.e., delay, cache hit rate, and throughput.

## 1. Introduction

In recent years, with increasing demands for high data rate applications, mass connectivity, universal internet access, Internet of Things (IoT), and wireless ad hoc networks, there has been an interest in LEO satellite mega-constellation networks, which are composed of multiple satellite orbits and hundreds of small satellites with lower delay [1,2,3,4]. Due to the LEO satellite mega-constellation supporting delay-sensitive services, it has become a hotspot for B5G and 6G [2,5,6,7,8]. What is more, the LEO satellite mega-constellation network for the IoT has become a common goal in the industry and academia [9,10,11,12]. On the other hand, the generated big data from IoT creates a severe challenge to the network architecture based on TCP/IP [13,14]. The IoT needs to support mobility, content distribution, and in-network caching. As a result, academic efforts have converged on a proposal for a future IoT architecture under the name of Information-Centric Networking (ICN) [15,16,17]. ICN is particularly valued in B5G and 6G core networks, WANET, IoT, and other fields [18,19,20,21]. Importantly, the architecture of ICN supports the high movement of satellite nodes [22], which causes the dynamic change of topology and intermittent connection between satellite nodes [19]. Meanwhile, it should be emphasized that the reasons why ICN supports the high movement of satellite nodes are because of ICN’s ubiquity of caching nodes, using caches not only at the edge but also at all levels of the network. Hence, in-network caching is an important feature, and how to manage cache is a key issue for the ICN network.

We explore the new concept of LEO satellite mega-constellation Information-Centric Networking for IoT, denoted by LEO-SMC-ICN-IoT in this paper and focus on in-network caching. In LEO-SMC-ICN-IoT, the LEO satellite mega-constellation not only functions as a network infrastructure, providing globally scalable connectivity with lower delay and cost, but also as a passive and host IoT access device for the physical world. Generally, the caching policy can be either proactive or reactive and solves the problem of how to cache while partially responding to when to cache [23]. In earlier proposals for ICN, the original caching scheme called the Leave Copy Everywhere (LCE) policy is described in [24,25]. For LCE, much cache space is wasted because the content is cached on every node in ICN, and lots of redundant content is produced. Furthermore, the Leave Copy Down (LCD) scheme is proposed in [24,26], the LCD caches the content at the downstream node of the hit node to improve the utilization of the cache space, but this approach is usually reactive.

The proactive approach predicts the popularity based on user preferences and pre-fetches content to further lower access time [23]. A proactive caching policy can considerably reduce the delay, increase the quality of experience, and relieve the burden on transit and backhaul links. So, the proactive policy is more suited for IoT services. Identifying how to make accurate predictions is the main issue for proactive caching approaches. To improve cache efficiency and performance, the paper [27] proposes a distributed and opportunistic (OPP) on-path caching scheme, where each on-path node independently picks contents to cache, and popular content is more likely to be cached by nodes. In [24], the contents are cached with a constant probability, and the policy is named random autonomous caching. The paper [28] proposes a heuristic caching policy based on probability, which combines the predicted content popularity and the content policy benefit. The paper [29] proposes a cooperative caching framework for mobile nodes caching data according to predicted preference. Unfortunately, the works above only make predictions of content, not nodes. Hence, while the locations of nodes are changing over time, the caching policy makes it difficult to accurately predict the popularity, which lowers caching efficiency. Furthermore, the policy with single content predication and time-variant characteristics is not suited for LEO-SMC-ICN-IoT.

The problems of proactive caching due to its time-variant characteristics can be extensively solved with Machine Learning (ML) [23,30,31]. ML can not only infer what to cache but also where to cache by predicting node locations [23,31,32,33]. So, we will apply ML to the proactive caching policy in LEO-SMC-ICN-IoT.

Random Forest (RF), a supervised ML algorithm for modeling high-dimensional data, can handle continuous, categorical, and binary data [34]. And the major features of RF that have gained attention are accurate predictions and better generalization due to the use of ensemble strategies and random sampling [35]. In this paper, we propose a novel caching decision policy using RF in LEO-SMC-ICN-IoT and focus on the caching policy of satellite nodes. So, the data of satellite nodes is taken as a resource. Also, the RF is introduced to predict the matching relationship between the satellite nodes and the content. Because of the optimization of the caching policy according to multi-dimensional data using attributes such as content popularity and satellite nodes’ caching rate, copies of content with high popularity are stored at locations close to the requesting satellite nodes. Hence, the LEO-SMC-ICN-IoT can provide content to Mobile Users (MU) with lower latency, i.e., better QoE. To our knowledge, past research on ICN caching policy has primarily focused on traditional or classical caching algorithms such as LCE and LCD [24,25,26], and some studies use improved ICN caching algorithms such as OPP, prob 0.5 [24,27]. While few studies on caching policies adapt to ML for different application scenarios, they do not consider satellite constellations [32,33]. Based on this, the contributions of this paper are listed as follows:(1)The framework of LEO-SMC-ICN-IoT is proposed. Using this framework, the caching policy can be dynamically managed, and adaptively optimize the caching performance.(2)We propose a novel caching algorithm of satellite nodes based on random forest denoted by CARF, which combines both content popularity prediction and satellite node location prediction. In CARF policy, using the concept “viscosity” [32,33], we present a cache matching rule: if the viscosity between content and satellite nodes is greater than some threshold, then the content is cached on satellite nodes.(3)Considering the transient and small characteristics of the IoT data [36], the proposed caching policy takes both the IoT data lifetime and the freshness as the important parameters and is more suited for IoT scenarios.

We organize the rest of this paper as follows: Section 2 introduces the LEO-SMC-ICN-IoT system model. In Section 3, we present the caching framework in LEO-SMC-ICN-IoT. In Section 4, we use the RF algorithm to design a proactive caching policy in LEO-SMC-ICN-IoT. Section 5 reports a performance evaluation of the proposed caching policy, by using the discrete event Network Simulator 3 (NS-3). Finally, we draw the conclusions.

## 2. System Model

In this section, the system model for the LEO-SMC-ICN-IoT is described. The model consists of the LEO-SMC-ICN-IoT scenario, LEO satellite mega-constellation network, and mobility calculation model.

### 2.1. LEO-SMC-ICN-IoT Scenario

The LEO-SMC-ICN-IoT scenario consists of a space segment and a ground segment. The space segment includes the LEO satellite mega-constellation network. And the ground segment includes the User Equipment (UE), the Base Station (BS), the Ground Station (GS), and the MU, etc. The contents can be cached at the satellite nodes, at the UE, at the BS, at the GS, or at the MU. The content caching scenario in LEO-SMC-ICN-IoT is shown in Figure 1. In Figure 1, the dashed line denotes the coverage of the satellite. Moreover, a solid line represents the IoT content caching and forwarding processes. The processes are briefly described as follows: The IoT content A is cached at GS, the IoT content B is cached at the satellite nodes, BS, and MU, and then the IoT content C is cached at UE. MU sends an interest A to BS. Since the IoT content A does not exist at BS, BS sends the interest A to GS and obtains IoT content A to return to MU. Compared with terrestrial communications, satellite communications have wide coverage, can realize the Internet of Everything, and can reduce the load on the ground segment. Here, to minimize user terminal content access delay, we consider the caching policy of satellite nodes for spatial segments, and the caching policy is applied to all satellite nodes. We present this in detail in Section 3.

### 2.2. LEO Satellite Mega-Constellation Network Mobility Calculation Model

Mobile technology has experienced exponential growth in the number of users, applications, and data traffic, with services ranging from multimedia streaming to online gaming [37,38]. Unfortunately, 5G technology is unable to meet the requirements of low latency and high bandwidth under IoT services. Fortunately, the satellite mega-constellation helps to meet stringent latency requirements [39,40].

We focus on the LEO satellite network and the orbital height in the range of 500~2000 km with a low transmission delay. Additionally, as a pioneer of LEO satellite mega-constellation networking, the Iridium constellation utilizes polar orbits to communicate with each other via inter-satellite links (ISLs) [41]. And the geometry structure of the Iridium constellation with six orbital planes (see Figure 2) can be simulated by the software SaVi 1.6.0 [42,43], the red line denotes the orbit of the satellite, the green dot denotes a satellite. In this paper, we use the Iridium constellation for analyzing the caching policy in LEO-SMC-ICN-IoT.

The LEO satellite mega-constellation is a complex mobility network. The prediction of the dynamic location of satellite nodes is a key point in caching policy. Our team has completed Section 2.2 of [41]. For ease of analysis, we briefly reiterate the orbit elements (also known as Keplerian elements) and the ISL geometry relationship of the Iridium constellation.

#### 2.2.1. Keplerian Elements

A satellite’s orbit can be described by seven orbital elements [44] (see Figure 3a): the semimajor axis α, eccentricity *e*, right ascension of the ascending node Ω, inclination δ, argument of the perigee ω, mean anomaly Μ, and true anomaly ν.

For ease of description, let the eccentric anomaly angle be E, the average angular velocity be n, and the time be t. Thus, the geometric relationship among M, v, and E (see Figure 3b) is given as:(1)M=nt,
(2)E−esinE=M=nt,
(3)$$v=2\tan^{-1}\sqrt{\frac{1+e}{1-e}\tan^{2}\frac{E}{2}},$$
(4)r=α(1−ecosE),
where r is the modulus of the satellite’s vector radius.

For the circular orbits commonly used in the LEO-SMC-ICN-IoT, e≈0 and M=E=v. Therefore, v is used to describe the Kepler elements of the constellation. The Keplerian elements of the Iridium constellation are listed in Table 1 [41].

#### 2.2.2. ISL Geometry Relationship of Iridium Constellation

Communication between any satellite nodes in the Iridium constellation exists in two ISL geometry relationships: two communicating satellite nodes on the same orbit (case 1) and two communicating satellite nodes on adjacent orbital planes (case 2) [41].

Case 1: ISLs between adjacent satellite nodes on the same orbital plane (see Figure 4).

In Figure 4, $r_{e}$ is the radius of the Earth, $r_{s}$ is the orbital height, θ is the angle between adjacent satellite nodes, and φ is the angle between $OS_{0-0}$ and $S_{0-0}S_{0-1}$, and then the distance of satellite node $S_{0-0}$ and node $S_{0-1}$ is:(5)$$d_{00-01}=\frac{\alpha \times \sin\theta }{\sin\varphi },$$

Case 2: ISLs between adjacent satellite nodes on adjacent orbital planes (see Figure 5).

The distance between $S_{0-0}$ and $S_{1-0}$ is:(6)$$d_{00-10}=\frac{\alpha \times \sin\theta /2}{\sin\varphi },$$

## 3. LEO-SMC-ICN-IoT Caching Framework

### 3.1. Mobility Prediction

With the high movement of satellite nodes in the LEO-SMC-ICN-IoT, the topology of the LEO satellite mega-constellation is high dynamics. However, the topology changes periodically as the satellite nodes move around periodically. Therefore, the mobility of the satellite nodes is predictable. Each satellite node follows an established orbit, returning to its initial position after one orbital period. The predictability of satellite nodes’ mobility can solve the problem of content caching due to satellite nodes’ movement.

According to Newton’s form of Kepler’s third law, the satellite orbit period *P* can be calculated using the following equation:(7)$$P=2\pi \sqrt{\frac{\alpha^{3}}{G\left(M_{e}+M_{s}\right)}},$$
where G represents the gravitational constant, $M_{e}$ is the earth mass, and $M_{s}$ is the satellite mass.

The handover management of satellite nodes is important because the satellite nodes’ high-speed movement, despite the handover management of satellite nodes, is not analyzed in this study. Even if the mobile terminal remains stationary on the ground, the mobile terminal must frequently hand over between satellites and beams, which makes the connection relationship between the terminal, the satellite nodes, and the GS relatively complex [45].

### 3.2. Content Popularity Prediction

In the LEO-SMC-ICN-IoT, the higher the popularity ranking, the higher the request probability. The accurate prediction of content popularity can improve cache performance [46]. Referring to the study of request popularity for terrestrial networks, the probability of a content being requested generally follows the Zipf distribution [47], i.e.,
(8)$$Z_{u}=\frac{{\left(u+q\right)}^{-\gamma }}{\sum_{u=1}^{U}{\left(u+q\right)}^{-\gamma }},$$
where *U* represents the total number of contents; *u* indicates the content popularity ranking, *u =* 1, 2, 3, …, *U*; $Z_{u}$ is the probability of requesting content with a content popularity rank of m; *q* is used to adjust the size of the content popularity rank; and γ is a parameter with values in [0.7, 1.3] [48]. The larger γ means that the set of very popular content in the content collection is relatively small.

### 3.3. ICN Forward Process

For the ICN Forward Process of the LEO satellite mega-constellation network, our team has completed Section 3.2 of [41]. For ease of understanding, we will briefly reiterate it.

The packet structure is shown in Figure 6 [49,50,51].

For performing the interest and data forwarding, each satellite node needs to maintain three functional components: Forwarding Information Base (FIB), Pending Interest Table (PIT), and Content Store (CS) [52]. Moreover, the interest and data forwarding steps in a satellite node are described in Figure 7 [41].

We set the GS_1_ as the interest requester and the GS_2_ as the data responder. In Figure 7 [41], the GS_1_ sends an interest with the name to a satellite node, and the satellite node checks whether a match data already exists in its local CS; if the responding data are found, then data is sent back to GS_1_ [41]. Otherwise, the satellite node investigates if an interest with the same name in the PIT already exists; if a similar interest is not found in the PIT, the new interest is forwarded according to the FIB and cached in the PIT [52]. When the GS_2_ receives the interest, data containing the requested content are sent back, and as the data arrives at a satellite node, the data are forwarded via ISLs, and the data are cached in the satellite node [41].

### 3.4. Caching Process in LEO-SMC-ICN-IoT

We consider a scenario of accessing the internet for ground terrestrial UT. Service providers are linked to the GS, which is responsible for distributing content via satellite mega-constellation networks. For UT content requests, the satellite nodes examine if the required content is currently cached in the local before forwarding the requested content to the next hop: if so, the satellite nodes send the requested content to the UT. Otherwise, the satellite nodes forward the requested content to the next hop. The caching process is shown in Figure 8. In Figure 8, the solid lines denote the connection of the two nodes to each other, and the dashed lines illustrate the content access route. Due to the consideration of the caching policy of satellite nodes only, we assume that all of the BS, and GS have no cached content at the beginning. The caching process is described as follows. Satellite $S_{0-1}$ has cached content $c_{1}$, and UT_1_ requests content; $c_{1}$ can access it via BS_1_ and GS_1_. In the case of UT_2_ requesting content $c_{1}$, it will access the requested content via ISLs from the satellite $S_{0-1}$. The other case is when UT_3_ accesses content $c_{7}$ from GS_n_ which is connected to the service provider. And $c_{L}\in C$, C is the set of contents.

Caching popular content in the LEO-SMC-ICN-IoT allows users with the same content requirements to be satisfied, without duplicate transmissions [52]. As a result, there is a significant decrease in redundant bandwidth and content access latency [53,54].

## 4. LEO Satellite Mega-Constellation Network Caching Policy

### 4.1. RF Algorithm

RF is one of the combined algorithms for data mining, reaching well-documented levels of accuracy and processing speed, and frequently appearing in new research [55]. RF as a supervised machine learning algorithm, uses a series of decision trees to distinguish between different classes of input data [35,56]. RF begins at the top of a decision tree and branches through a series of binary decisions, eventually leading to the input sample being defined as one of the possible categories [35], as shown in Figure 9. The process of establishing a random forest is as follows:

Step 1: The sample set is sampled *n* times to form a new sample set.

Step 2: Feature subsets are built by randomly selecting features (e.g., cache rate, replacement rate, and popularity).

Step 3: On the new sample set and feature subset, the best segmentation attribute is found, and the decision tree is established.

Step 4: *H* decision trees are constructed by repeating Steps 1–3 with *m* times.

Step 5: The random forest model is built by combining the results of *H* decision trees.

### 4.2. Caching Based on IoT Data Lifetime

The concepts of “IoT content lifetime”, and “IoT content freshness” need to be defined first, because the IoT data are typically transient and small. Then, the caching policy based on RF is described.

IoT content lifetime (CLT) is defined as the time length between when data is generated by a content producer and when it is no longer valid, i.e., expires [57]. In the packet of ICN, there is a signature information domain containing information such as publisher ID, key locator, stale time, timestamp, etc. [57]. A satellite node learns when the data are generated through a content producer by checking the timestamp.

IoT content freshness f is denoted by [58]:(9)$$f=T_{c}-T_{g},$$
where $T_{c}$ is the current time, and $T_{g}$ is the time when the data are generated.

The data freshness is zero when $T_{c}=T_{g}$. When f=CLT, the content should be discarded.

The same content may have different freshness requirements for different applications. Content may only be sent back to an application if the freshness value of the content is less than the freshness requirement of the application.

In this study, the IoT devices, e.g., the radars, file servers, and IoT sensors, etc., are the content producers, and the content consumers are the UT.

The UT sends interests to retrieve content. This paper denotes the IoT-requested content at time *t* by $Req(t)=\left\{req_{1}(t),req_{2}(t),req_{3}(t),\ldots ,req_{n}(t),\ldots ,req_{N}(t)\right\}$. Each request is composed of $req_{n}(t)=\langle c_{l},f_{c_{l}},t\rangle$, where $c_{l}\in C$, $C=\left\{c_{1},c_{2},c_{3},\ldots ,c_{l},\ldots ,c_{L}\right\}$ is the set of contents, $f_{c_{l}}$ is the freshness of contents $c_{l}$, and t is the time at which the request arrives.

All the satellite nodes have caching capability to cache content. s(i,t)∈{0,1} is used to denote the caching status of the satellite node i at time t. If the content $c_{l}$ is cached at the satellite node i at the time t, s(i,t)=1; otherwise,s(i,t)=0. Furthermore, $R(Req(t),\sum_{i=1}^{\left|I\right|}s(i,t))=\left\{r_{1}(t),r_{2}(t),r_{3}(t),\ldots ,r_{n}(t),\ldots ,r_{N}(t)\right\}$ is used to indicate how many requests in the requests set Req(t) that can be obtained from the satellite node i, where i∈I, I is the satellite nodes set, |I| is the total number of satellite nodes. $r_{n}(t)\in \left\{0,1\right\}$ means whether the $n^{th}$ request can be served from the satellite nodes at a time t. By applying a caching policy CARF, $s(i,t)\overset{CARF}{\to }s(i,t+\Delta t)$, the new caching status of the satellite node i at time t+Δt can be achieved.

### 4.3. Caching Policy Based on RF

Due to the dynamic characteristics of the satellite nodes, the content required by the CARF needs to be able to describe both satellite nodes’ location and content properties. Thus, the policy is analyzed in terms of both the satellite nodes and content dimensions.

The satellite node dimension is utilized to represent the load status of a satellite node during different time intervals. Therefore, we give the definitions of the cache rate and the cache replacement rate.

**Definition 1.** *The cache rate is defined as the load rate during the unfilled phase of the cache load for satellite nodes, and the cache rate is given by:*(10)$$CR=\frac{\sum_{c=1}^{L}CS_{c}}{CZ},$$*where* $CS_{c}$ *is the size of the cached contents, * L *represents the amount of cached content* c *within unit time, and* CZ *represents the cache capacity of the satellite node i.*

**Definition 2.** *The cache replacement rate is given by:*(11)$$RR=\frac{\sum_{c=1}^{\bar{L}}RS_{c}}{CZ},$$*where* $RS_{c}$ *is the replaced content size,* $\bar{L}$ *represents how many replaced content c within unit time. The replacement rate is used to define the operating state of the satellite nodes’ cache capacity during the full load phase. If the network is in a stable status and the cache space of some satellite nodes is occupied, the cache replacement rate can describe the load and cache status of the satellite nodes effectively, which reflects the timeliness of different contents at the satellite nodes [58]*.

Content popularity goes through a dynamic process of rising, peaking, and finally decaying in popularity [46]. Content popularity is also affected by location and request rate [46]. Thus, even with the same content, the popularity may not be the same on different satellite nodes. In terms of content dimension, the temporal relevance and spatial correlation of popularity are discussed here. In terms of temporal relevance, popularity describes the dynamic trend in the number of content requests. From a spatial association perspective, viscosity is adopted to define matching relationships between the content of different popularity and satellite nodes at different locations.

Now, we give the definitions of the content popularity, the content request weight, and the cache viscosity.

**Definition 3.** *The content popularity is given by:*(12)$$CP=\frac{RN}{\sum_{c=1}^{L}TR_{c}},$$*where* RN *is how much content* c *requests within unit time according to the PIT table,* L *represents how much content* c*, and* $\sum_{c=1}^{L}TR_{c}$ *represents the total number of requests in the satellite node* i 
*within a unit of time.*

**Definition 4.** *The content request weight is given by:*(13)RW=logℕM,*where* M *represents how many satellite nodes send the requested content* c*, and* ℕ *indicates the total number of satellite nodes.* logℕ/M *represents a global factor of all satellite nodes and performs the relative importance of requested content* c *to all satellite nodes, without considering any specific satellite nodes. The value becomes smaller when more satellite nodes send the same content, meaning that there is less correlation between the request and the satellite nodes. RW is proportional to the popularity of the content on the satellite nodes and inversely proportional to the number of requesting satellite nodes in the whole network.*

The matching relationship value checks whether the content c matches the satellite node i and is defined as follows:

**Definition 5.** *The cache viscosity is given by:*(14)CV=RP×logℕM¯,*where* M¯ *represents how many satellite nodes cache content* c*,* RP=CT/∑c=1LCNc *is the caching response rate,* CT *represents response time to cached content* c *in the current satellite node* i *within unit time according to the CS table, and* ∑c=1LCNc *counts the total number of requests to all cached contents in the satellite node* i.

Using the cache viscosity, the cache matching rule is given by:

Matching rule: The matching relationship between cached content and satellite nodes is given by calculating the cache viscosity. The higher the viscosity value of the cache, the greater the matching relationship between the satellite nodes and the content, and the satellite nodes are more suited to the content. The threshold value of a matching relationship is given by ε; if the cache viscosity CV≥ε, this means there is a good match with the content c on the satellite node i, and content c is suited to be cached in the satellite node i, then s(i,t)=1; otherwise, s(i,t)=0. The matching rule guides the deployment of caches. Especially, the content is cached in advance at the future satellite to maintain communication with the current ground segment at the time of satellite switchover.

The caching Algorithm 1 flow diagram is shown in Figure 10.

The cache policy CARF overall process is as follows.
**Algorithm 1:** Caching algorithmInput: the state attribute value CR, RR, CP, RW, and CV, the threshold value ε.Output: the caching status at the moment t+Δt.1: Set *L* is the number of all contents c; I is the number of all satellite nodes i; The input X={CR,RR,CP,RW}; Set caching status s(i,t)=0; Set T is the time;2: if T=t then3:     for c=1 to *L* do4:        if CV≥ε, then caching status s(i,t)=1;5:        else s(i,t)=0;6:       end if7:      end for8: end if9: if T=t+Δt then10:   for c=l+1 to *L* do11:           solving by the CARF caching policy, calculating the cache viscosity, if 12:            CV≥ε, s(i,t+Δt)=1, then cache the content c in the satellite node i;13:      else14:             do not cache the content c in the satellite node i;15:      end if16:   end for17: end if

## 5. Simulation Experiments and Analysis

### 5.1. Design of LEO-SMC-ICN-IoT Simulation Platform

For the design of the LEO satellite mega-constellation network simulation platform, our team has completed Section 4.1 of [41]. For ease of understanding, we will briefly reiterate it.

The NS-3, a discrete event-driven network simulator, consists of interdependent modules; the ndnSIM module is the latest ICN simulation module based on NS-3, which implements the basic components of ICN modularly [12]. Yet the current ndnSIM can only provide the lower layer support of the ICN protocol stack, such as CS, PIT, and FIB, forwarding policy, and cache replacement strategy [59]. Therefore, this paper expands the NS-3 structure by adding LEO modules and develops a hierarchical simulation platform, as shown in Figure 11 [41]; the simulation platform consists of four layers: infrastructure, a network model that considers the number of orbital planes in the LEO-SMC-ICN-IoT network, the number of satellite nodes in each orbital plane, the orbital height of the satellite constellation, and ISLs to construct a mobile model with mobility prediction [41]; a network configuration configures the broadcast channel for the LEO-SMC-ICN-IoT; the control layer is responsible for maintaining and updating the ISLs and satellite-to-ground links, and is also responsible for data packet caching and routing between satellite nodes in the network.

### 5.2. Simulation Scene

According to the Iridium constellation in Figure 2, some parameters are configured to maintain dynamic ISL updating, and the parameters are listed in Table 1 [41].

The coordinates of the two GS on Earth are (17° S, 31° E) and (48° N, 99° E), and are denoted by GS_1_ and GS_2,_ respectively, as shown in Figure 12. GS_1_ is the requester of interest, and GS_2_ is the data responder. Communication between the two GSs is established via the ISLs with multiple hops [41].

The content forward process is described as follows: When the GS_1_ requests content, the connecting satellite nodes will first check if a copy of the content is stored in the local cache. If a copy of the content is already cached, the satellite nodes will return the content directly in response to the request. If no copy is cached, the satellite nodes forward the requested packet of interest to the surrounding satellite nodes via ISLs, and the surrounding satellite nodes will be in response to the request. If the content is not available in the entire satellite network, the request will be forwarded to the GS_2_.

### 5.3. Simulation Setup

IoT services include sensor data, RFID data, image and video, etc. And the simulation parameters settings are listed in Table 2.

The simulations compare the performance of our proposed policy with that of the LCE [24,25], the OPP [27], the LCD [24,25], and the probabilistic algorithm which caches each content with probability 0.5 (prob 0.5) [28]. The request pattern follows a Zipf distribution, and the request process for content in the network exhibits a poison distribution. The contents lifetime is uniformly spread between one second and one minute. It is assumed that each content cache occupies one cache unit and no content is cached at each satellite node in the initial state. And the default broadcast routing policy is used.

### 5.4. Performance Analysis

The cache hit ratio is an essential metric for assessing the efficiency of a caching policy and is defined as the percentage of requests that can be satisfied by the LEO-SMC-ICN-IoT caching system. Therefore, the cache hit ratio is denoted by:(15)$$Hit_{CARF}=\sum_{t=1}^{T}\frac{1}{|Req(t)|}\cdot R(Req(t),\sum_{t=1}^{T}s(i,t)),$$
where *T* is the total time, $R(Req(t),\sum_{t=1}^{T}s(i,t))$ is the number of requests that hit the satellite node at time t, and |Req(t)| represents the total number of requests at time t.

The content retrieval delay is measured in terms of the average number of hops, which is calculated as:(16)$$Hop_{CARF}=\sum_{t=1}^{T}\frac{1}{|Req(t)|}\cdot Hop(Req(t),\sum_{t=1}^{\left|I\right|}s(i,t)),$$
where $Hop(Req(t),\sum_{t=1}^{\left|I\right|}s(i,t))=\{hop_{1}(t),hop_{2}(t),\ldots ,hop_{n}(t),\ldots ,hop_{N}(t)\}$ is the number of hops used to satisfy each request at time t. If $req_{n}(t)$ can be met from the satellite node i, $hop_{n}(t)$ is equivalent to the amount of hops from the satellite node i to the user; otherwise, $hop_{n}(t)$ equals the amount of hops from the content producer to the user.

We analyze the cache capacity and Zipf distribution on the performance of the LEO-SMC-ICN-IoT. And the request rate of users is assumed to be 100 packets of interest per second.

The cache performance such as the average number of hops, average hit rate, and throughput by varying the cache size is shown in Figure 14. The cache size is the ratio caching capacity of the satellite node to the total contents in the LEO-SMC-ICN-IoT. Figure 13 shows that the hit rate and the throughput using the five caching policies gradually increase with the cache size increasing. And the average number of hops gradually decreases. This result occurs because the amount of content that can be cached in the satellite nodes increases at the time t+Δt with the cache size increasing. These results lead to an increased probability. A comparative analysis shows that CARF performs better than the other four caching strategies in three evaluation metrics, including average hop count, hit rate, and throughput. This is because the LCE, prob 0.5, and LCD cache content are redundant, which results in the cached content being less diverse. In addition, because OPP uses the relationship between the popularity of the content and the location of satellite nodes, the cache performance is better than the LCE, LCD, and prob 0.5. Considering the relationship between the popularity of the content and the location of satellite nodes, the CARF policy further improves the utilization of cache space and service efficiency by learning from satellite nodes’ location and content using RF.

The Zipf distribution parameter γ mainly reflects the degree of concentration of user preferences. We vary γ to investigate the impact of cache performance, and the trend of different cache performance is shown in Figure 14. As the γ increases, the average hit rate and throughput of the five caching policies gradually increase, and the average number of hops gradually decreases. The reason is that with the γ increasing, there is an increasing request for hot content in the LEO-SMC-ICN-IoT network. The LCE, the LCD, and the prob 0.5 have a little performance improvement when the popularity increases, while they are insensitive to changes in content popularity. Finally, because the popular content using CARF is more concentrated than the OPP, the CARF outperforms the OPP.

The request rate is the number of interest packets sent by a satellite node per unit time. We set the cache size to 40% and the Zipf parameter to 1. Figure 15 describes the trend for the average number of hops under different request rates. The performance metrics for each cache policy did not change significantly as the request rate changed, which shows that the current request rate is within the capability of the satellite nodes. The proposed CARF outperforms other caching approaches for all different request rates, and compared to the OPP, it can reduce the average number of hops by around 15%. This means that the proposed CARF can reduce the IoT devices’ energy consumption and the data retrieval delay.

## 6. Conclusions

In this paper, we analyze the advantage of the ICN paradigm in IoT scenarios for seamless global coverage. Thus, the framework of the LEO-SMC-ICN-IoT is proposed first. The framework combines LEO satellite mega-constellation, ICN, and IoT. Based on the LEO-SMC-ICN-IoT, we focus on the in-network caching policy for satellite nodes:(1)Introducing the concept of “viscosity”, the proposed CARF caching policy of satellite nodes combines both content popularity prediction and satellite nodes’ location prediction, for achieving good cache matching between the satellite nodes and content.(2)Using the RF algorithm which considers the IoT content lifetime and freshness, the policy can accurately predict the matching relationship between the satellite nodes and the content.

Developing an LEO-SMC-ICN-IoT simulation platform by NS3, the proposed CARF caching policy performs markedly better in terms of the average number of hops, hit rate, and throughput, which also improves the quality of user experience. We conclude that caching performance can be significantly improved by fully analyzing and exploiting the matching relationship between the satellite nodes and the content using RF.

## Figures and Tables

**Figure 1 sensors-24-03412-f001:**
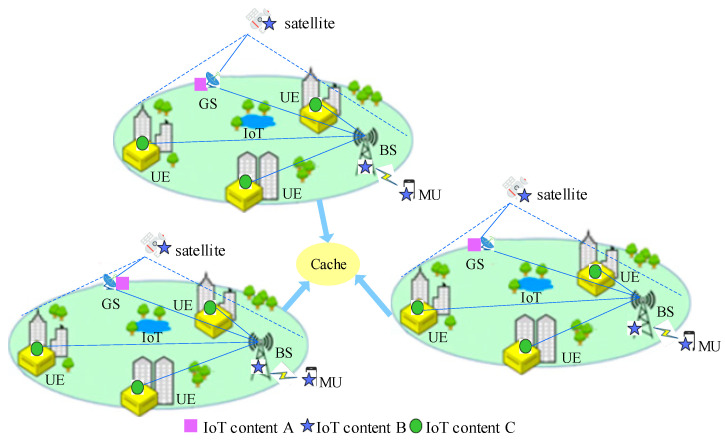
LEO-SMC-ICN-IoT scenario.

**Figure 2 sensors-24-03412-f002:**
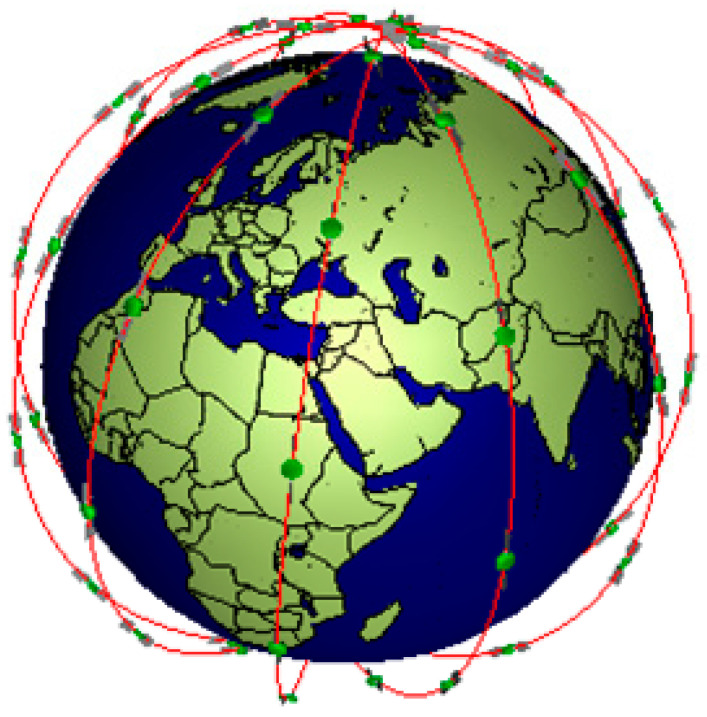
Iridium constellation.

**Figure 3 sensors-24-03412-f003:**
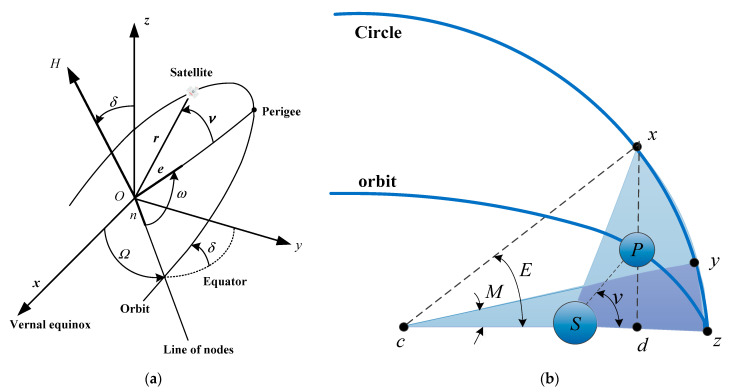
Schematic diagram: (**a**) Keplerian elements; (**b**) Geometric relationship among *M*, v, and *E*.

**Figure 4 sensors-24-03412-f004:**
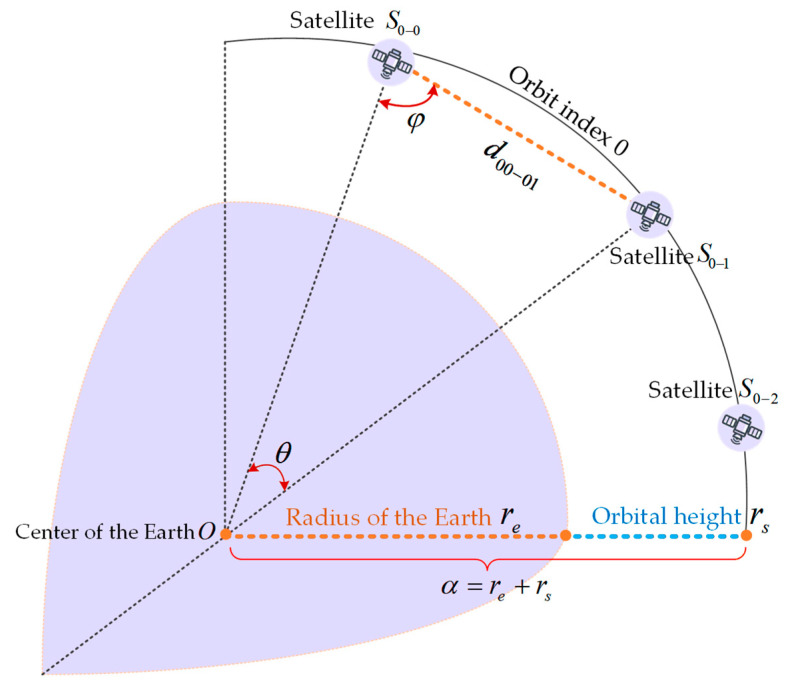
ISLs between two satellite nodes on the same orbit.

**Figure 5 sensors-24-03412-f005:**
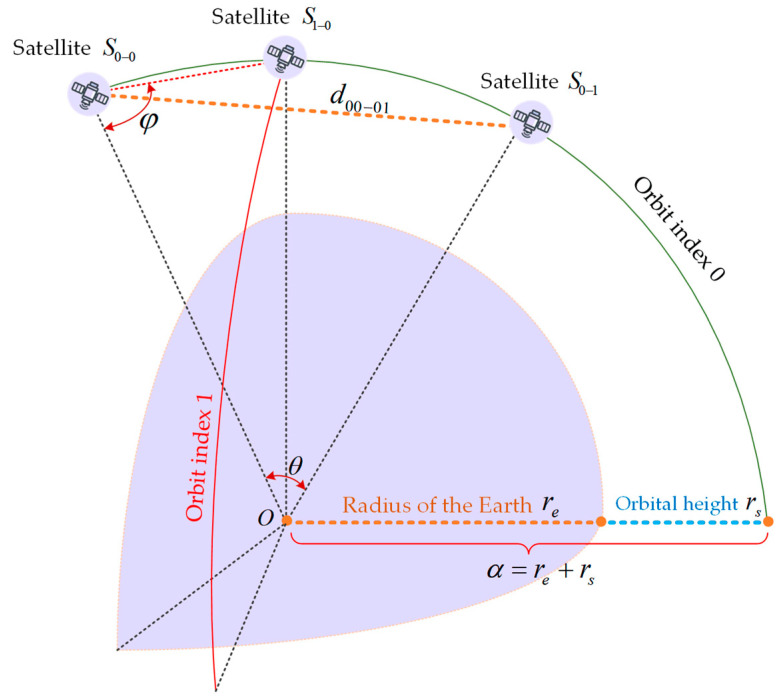
ISLs have the closest communication distance between adjacent orbital planes.

**Figure 6 sensors-24-03412-f006:**
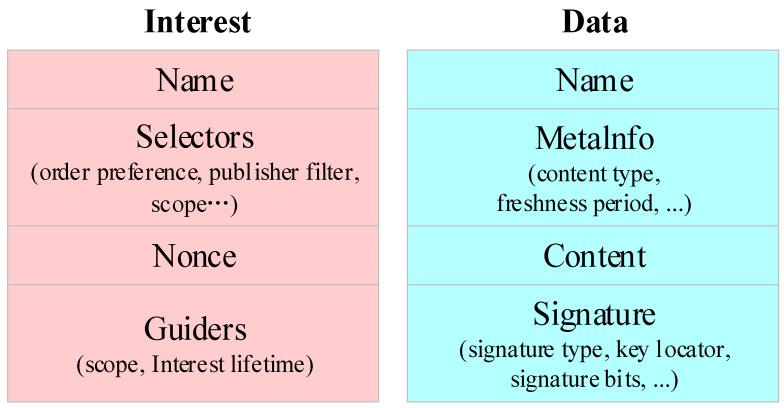
Packet architecture.

**Figure 7 sensors-24-03412-f007:**
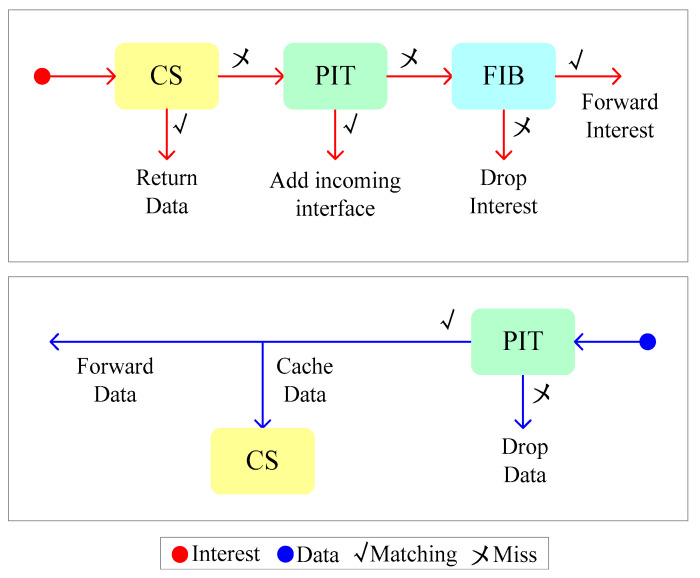
Forwarding process at a satellite node.

**Figure 8 sensors-24-03412-f008:**
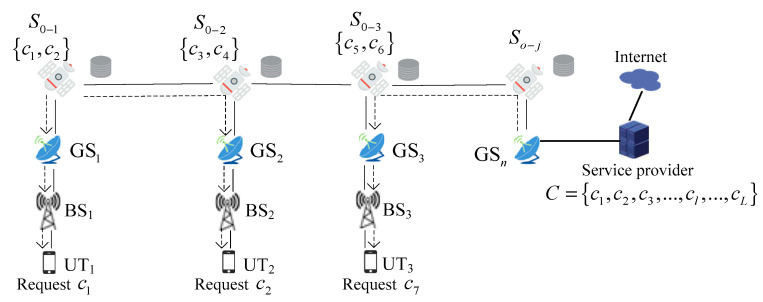
Caching process in LEO satellite mega-constellation networks.

**Figure 9 sensors-24-03412-f009:**
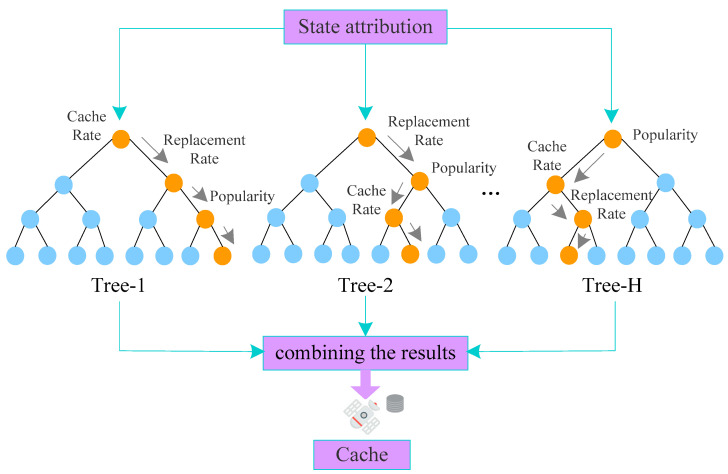
RF algorithm.

**Figure 10 sensors-24-03412-f010:**
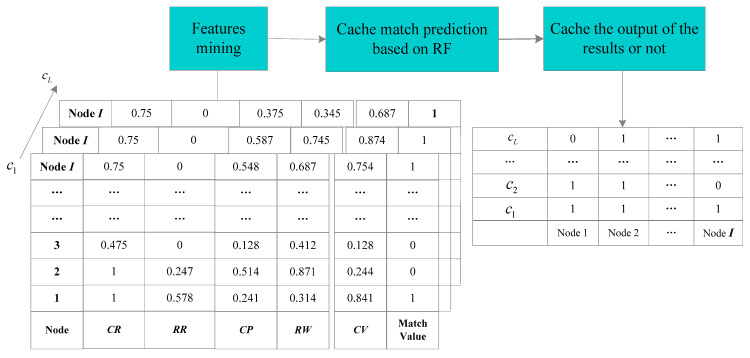
Caching algorithm flow diagram.

**Figure 11 sensors-24-03412-f011:**
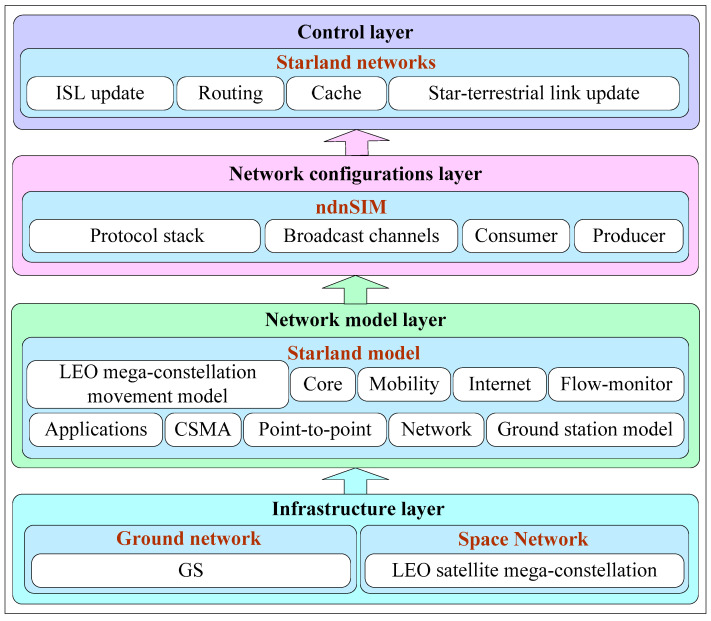
The architecture of the LEO-SMC-ICN-IoT simulation platform.

**Figure 12 sensors-24-03412-f012:**
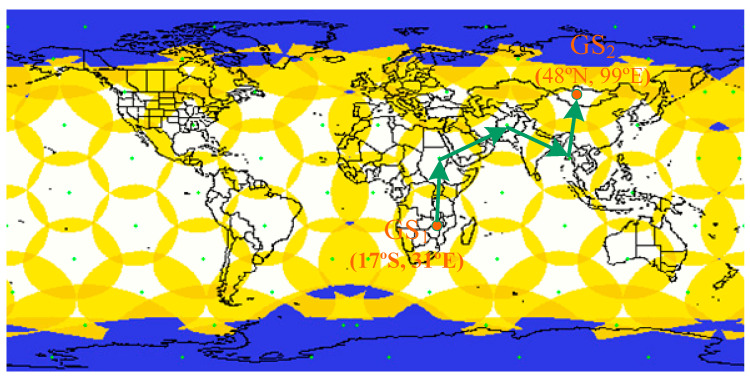
Two-dimensional LEO satellite constellation with ISL coverage diagram.

**Figure 13 sensors-24-03412-f013:**
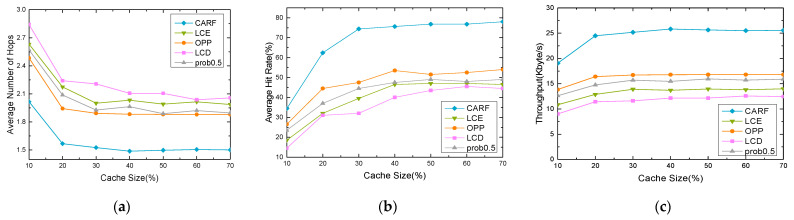
Impact of cache size: (**a**) Average content access delay; (**b**) Average hit rate; (**c**) Throughput.

**Figure 14 sensors-24-03412-f014:**
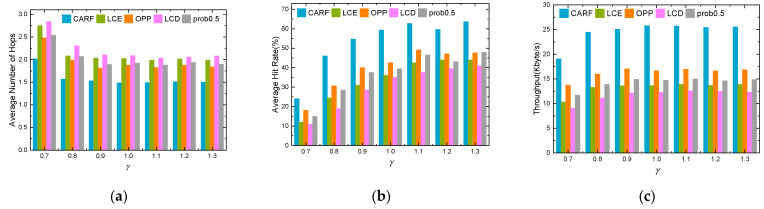
Impact of γ: (**a**) Average content access delay; (**b**) Average hit rate; (**c**) Throughput.

**Figure 15 sensors-24-03412-f015:**
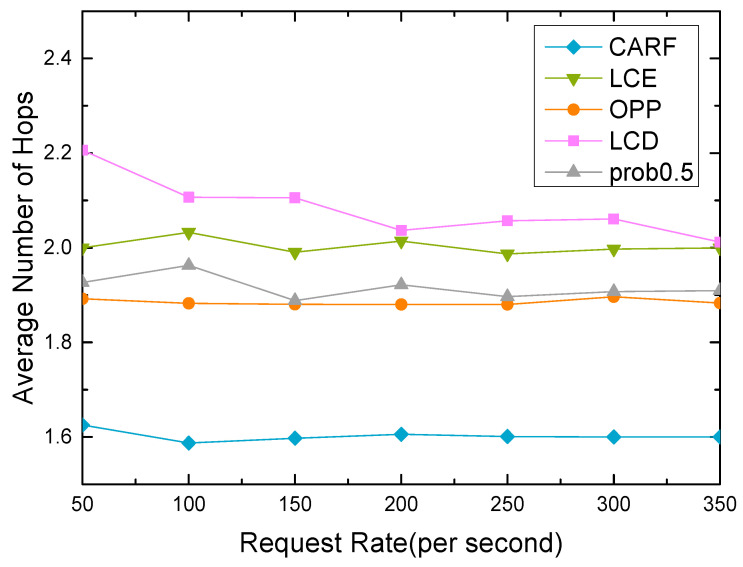
The trend of the average hop rate with the request rate.

**Table 1 sensors-24-03412-t001:** Keplerian elements of the Iridium constellation.

Element	Symbol	Value	Comment
Semimajor axis (km)	α	7159.14	Earth radius + orbital altitude
Eccentricity	*e*	0.0002	Circular orbit
Right ascension of ascending node (degrees)	Ω	0, 31.80, 63.60, 95.41, 127.21, 159.01	Six orbital planes
Inclination (degrees)	δ	86	Non
Argument of perigee (degrees)	ω	0	Circular orbit
True anomaly (degrees)	v	0, 32.73, 65.45, 98.18, 130.91, 163.64, 196.36, 229.09, 261.82, 294.55, 327.29	11 satellite nodes evenly distributed in odd plane
		16.36, 49.09, 81.82, 114.55, 147.27, 180, 212.73, 245.45, 278.18, 310.91, 343.64	11 satellite nodes evenly distributed in even plane

**Table 2 sensors-24-03412-t002:** Parameter Settings.

Description	Value	Description	Value
Constellation	Iridium	ISL bandwidth	25 Mbps [60]
Number of satellite nodes	66	Star-ground link bandwidth	1.5 Mbps [60]
Number of GS	2	Simulation time	200 s
Number of contents	2000	Replacing policy	Least Recent Used (LRU) [61]
Packed size	500 byte [48]		

## Data Availability

Data are contained within the article.

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
