# Peer review of "Caching Policy in Low Earth Orbit Satellite Mega-Constellation Information-Centric Networking for Internet of Things"

_sensors, 2024, doi:10.3390/s24113412_

Round 1

Reviewer 1 Report

Comments and Suggestions for Authors

-The delay of the network should be taken into account in detail.
-Comparison between the proposed method, and some other well-known methods existing in the literature should be given.
-The ref section should be updated, and the latest refs related to the topic should be given.
-If it is possible, and there is no conflict-of-interest, I kindly suggest the authors to open-source their code to disseminate their findings, or at least some portion of the code.

-Some figures are hard to read, their size should be increased.

-The paper compares different algorithms populaly being used in the topic, CARF, RF, LCE, OPP, LCD. I kindly suggest the authors why they chose to compare these methods, not others, what makes these algorithms valuable to compare ?

-The authors introduce the concept of LEO satellite mega-constellation information-centric networking for IoT, referred to as LEO-SMC-ICN-IoT in this paper, with a focus on in-network caching. In LEO-SMC-ICN-IoT, the LEO satellite mega-constellation serves not only as a network infrastructure, providing globally scalable connectivity with lower latency and cost, but also as a passive and host IoT access device for the physical world.

Traditionally, caching policies can be either proactive or reactive, addressing the question of both what to cache and when to cache. In earlier proposals for ICN, the original caching scheme known as Leave Copy Everywhere (LCE) policy was introduced. LCE leads to significant cache space wastage as content is cached on every node in the ICN, resulting in a large amount of redundant content. Additionally, the Leave Copy down (LCD) scheme, caches content at the downstream node of the hit node to improve cache space utilization, although this approach is typically reactive.

-The number of the references should be increased though.

-The quality and resolution of the images/figures are well enough. Regarding the figures, I would kindly suggest the authors to share the data-sets, if there is no conflict of interest, to disseminate the knowledge among people working in the similar fields.

Comments on the Quality of English Language

There are a few minor English grammar mistakes, these should be fixed.

Reviewer 2 Report

Comments and Suggestions for Authors
